# Cannabinoids versus placebo for pain: A systematic review with meta-analysis and Trial Sequential Analysis

Jehad Barakji[1]*, Steven Kwasi Korang[1], Joshua Feinberg[1,2], Mathias Maagaard[1,3], Ole Mathiesen[3,4], Christian Gluud[1,5], Janus Christian Jakobsen[1,5]

1 Copenhagen Trial Unit, Centre for Clinical Intervention Research, The Capital Region, Copenhagen University Hospital — Rigshospitalet, Copenhagen, Denmark, 2 Medical Department, Cardiology Section, Holbaek University Hospital, Holbaek, Denmark, 3 Centre for Anaesthesiological Research, Department of Anaesthesiology, Zealand University Hospital, Køge, Denmark, 4 Department of Clinical Medicine, Copenhagen University, Copenhagen, Denmark, 5 Department of Regional Health Research, The Faculty of Heath Sciences, University of Southern Denmark, Odense, Denmark

* jehad.barakji@ctu.dk

## Abstract

### Objectives

To assess the benefits and harms of cannabinoids in participants with pain.

### Design

Systematic review of randomised clinical trials with meta-analysis, Trial Sequential Analysis, and the Grading of Recommendations Assessment, Development and Evaluation (GRADE) approach.

### Data sources

The Cochrane Library, MEDLINE, Embase, Science Citation Index, and BIOSIS.

### Eligibility criteria for selecting studies

Published and unpublished randomised clinical trials comparing cannabinoids versus placebo in participants with any type of pain.

### Main outcome measures

All-cause mortality, pain, adverse events, quality of life, cannabinoid dependence, psychosis, and quality of sleep.

### Results

We included 65 randomised placebo-controlled clinical trials enrolling 7017 participants. Fifty-nine of the trials and all outcome results were at high risk of bias. Meta-analysis and Trial Sequential Analysis showed no evidence of a difference between cannabinoids versus placebo on all-cause mortality (RR 1.20; 98% CI 0.85 to 1.67; P = 0.22). Meta-analyses and Trial Sequential Analysis showed that cannabinoids neither reduced acute pain (mean

**Funding:** Funding JB received a grant for this research from A.V. Lykfeldt and Wife's Grant (A.V. Lykfeldt og Hustrus Legat).

**Competing interests:** The authors have declared that no competing interests exist.

difference numerical rating scale (NRS) 0.52; 98% CI -0.40 to 1.43; P = 0.19) or cancer pain (mean difference NRS -0.13; 98% CI -0.33 to 0.06; P = 0.1) nor improved quality of life (mean difference -1.38; 98% CI -11.81 to 9.04; P = 0.33). Meta-analyses and Trial Sequential Analysis showed that cannabinoids reduced chronic pain (mean difference NRS -0.43; 98% CI -0.72 to -0.15; P = 0.0004) and improved quality of sleep (mean difference -0.42; 95% CI -0.65 to -0.20; P = 0.0003). However, both effect sizes were below our predefined minimal important differences. Meta-analysis and Trial Sequential Analysis indicated that cannabinoids increased the risk of non-serious adverse events (RR 1.20; 95% CI 1.15 to 1.25; P < 0.001) but not serious adverse events (RR 1.18; 98% CI 0.95 to 1.45; P = 0.07). None of the included trials reported on cannabinoid dependence or psychosis.

## Conclusions

Cannabinoids reduced chronic pain and improved quality of sleep, but the effect sizes are of questionable importance. Cannabinoids had no effects on acute pain or cancer pain and increased the risks of non-serious adverse events. The harmful effects of cannabinoids for pain seem to outweigh the potential benefits.

## Introduction

Pain is the most commonly reported symptom in the general population and medical settings [1–3]. Persistent or chronic pain is a major universal health problem [4], prompting the WHO to endorse a global campaign against pain [5]. Pain has been associated with lower degree of health-related quality of life and may lead to psychosocial distress, insomnia, and depressive symptoms [6–14].

### Cannabinoids

Cannabinoids have lately emerged as a potential alternative to other analgesics, e.g., opioids for the treatment of pain [15]. Cannabinoids are most commonly consumed via smoked, inhaled vapour, or oral routes of administration [16]. Sublingual administration is used for some medical cannabis preparations (e.g., nabiximol).

The endocannabinoid system consists of two types of cannabinoid receptors in the human body, type I and type II [17]. Cannabinoid receptor type I are most abundant in the central nervous system, especially in areas stimulating nociception and short-term memory, and in the basal ganglia. Cannabinoid receptor type II is mostly found in the periphery, often in conjunction with immune cells, but may appear in the central nervous system predominantly in association with microcytes during inflammation [17].

These receptors is thought to suppress the pain stimulus through different mechanisms [18]. Neurochemical, behavioral, and electrophysiological studies all demonstrated the modulation of inflammatory nociception through cannabinoid receptor type II activation [19].

### Pain

Acute pain usually has a well-defined onset and most often a readily identifiable etiology (i.e., surgery, etc.). Acute pain is expected to run its course in a relatively short time frame and management of acute pain typically focuses on providing symptomatic relief until pain is reduced to an acceptable level [20].

Chronic pain is typically defined as pain lasting for more than three months [21]. Chronic pain may also have a well-defined onset related to tissue injury (e.g., surgery) and be mediated through an intact nervous system. It may, however, also be caused by nerve damage and dynamic changes in the nervous system, and be characterized by an ill-defined onset and a prolonged, fluctuating course [20].

Pain may also be classified based on whether it is cancer-related or non-cancer-related. Cancer-related pain is caused by the cancer itself (primary tumor and metastases) or its treatment (e.g., radiation therapy) [22].

Previous reviews have reported on serious adverse events (e.g., agitation, impaired memory, abuse, dissociation, acute psychosis, and death) [23–27] and non-serious adverse events (e.g., sedation, dizziness, dry mouth, increased appetite, somnolence, confusion and nausea) [23–25, 27–31] in users of cannabinoids for pain.

According to Canada's Drug and Health Technology Agency (CADTH) there is some suggestion of benefit with cannabis-based medicines specifically for neuropathic pain. However, such benefits needs to be weighed against harms [32].

Before healthcare systems ought to endorse the applicability of cannabinoids for pain globally, the potential short- and long-term benefits and harms of cannabinoids must be investigated. We conducted this systematic review with meta-analysis and Trial Sequential Analysis (TSA) and based on available randomised trials to evaluate the effectiveness and safety of cannabinoids in participants with pain.

## Methods

The objective of our systematic review was to assess the benefits and harms of cannabinoids versus placebo or no intervention for participants with any type of pain (acute and chronic pain, cancer pain, or any other types of pain). Our methodology is described in detail in our protocol published prior to conducting the literature search [33].

In short, we carried out this systematic review following the recommendations of the Preferred Reporting Items for Systematic Reviews and Meta-Analysis (PRISMA) guidelines [34]. We included all randomised clinical trials comparing cannabinoids versus placebo or no intervention for participants with any type of pain. Two authors (JB, SKK) independently searched for trials identified prior to January 2022 see '**Supplement 1 in S1 File**' for a detailed list of databases and '**Supplement 2 in S1 File**' for the search strategy. We included randomised clinical trials regardless of trial design, setting, publication status, year, language, and reporting of outcomes. Four authors (JB, SKK, JBF, and MM) working in pairs independently extracted data and assessed the risks of bias in included trials. We contacted trial authors by email if data were unclear or missing. Disagreements were resolved through discussion or by consulting a third author (JCJ).

### Outcomes

**Primary outcomes.**

- All-cause mortality

- Pain assessment on visual analogue scale (VAS) or numerical rating scale (NRS)

- Proportion of participants with one or more serious adverse events. Serious adverse events were defined as any untoward medical occurrence that resulted in death; was life threatening; was persistent or led to significant disability, hospitalisation, or prolonged hospitalisation [35]. As we expected the trialists' reporting of serious adverse events to be heterogeneous and not strictly according to the International Conference on Harmonisation

—Good Clinical Practice (ICH-GCP) recommendations, we included the event as a serious adverse if the trialists either: (1) used the term 'serious adverse event' but did not refer to ICH-GCP, or (2) reported the proportion of participants with an event we considered fulfils the ICH-GCP definition. If several of such events were reported then we choose the highest proportion reported in each trial

- Quality of life measured on any valid (published validation) continuous scale

**Secondary outcomes.**

- Cannabinoid dependence (as defined by trialists)

- Psychosis (as defined by trialists)

- Proportion of participants with one or more adverse event not considered to be serious

- Quality of sleep measured on any valid (published validation) continuous scale

**Exploratory outcomes.**

- Each type of serious adverse event separately

- Each type of adverse event not considered serious separately

- Twenty-four hours morphine consumption (as defined by trialists)

- Physical function (as defined by trialists)

- Depressive symptoms (e.g., Hamilton Depression Rating Scale)

Adverse events were included in our analysis regardless of whether it was defined as an outcome by the trialists.

For all outcomes, we used the trial results reported at maximal follow-up.

**Patient and public involvement.** We conducted email correspondence with several patient associations in Denmark to select the most patient-relevant outcomes. The associations were the Danish Diabetes Association, the Danish Rheumatism Association, the Danish Multiple Sclerosis Society, and the Danish Cancer Society.

**Sub-group analyses.** We pre-defined subgroup analyses for our primary outcomes assessing risk of bias, risk of vested interests, type of pain, type of chronic pain, and type of cannabinoids used [33].

**Assessment of risk of bias.** We assessed risk of bias according to the Cochrane Handbook for Systematic Reviews of Interventions 5.1 [36]. We evaluated the risk of bias in the domains 'random sequence generation', 'allocation concealment', 'blinding of participants and treatment providers', 'blinding of outcome assessment, 'incomplete outcome data', and 'selective outcome reporting'. We used these domains to classify the included trials as being at overall low risk of bias or at overall high risk of bias as described in our protocol [33].

The domains 'blinding of outcome assessment', 'incomplete outcome data', and 'selective outcome reporting' were further assessed separately for each outcome result.

**Assessment of effect.** We calculated risk ratios (RRs) with 95% and 98% confidence intervals (CIs) for our continuous and dichotomous outcomes (see "**Assessment of statistical and clinical significance"**). For our dichotomous outcomes we used the conventional direction with RR > 1.0 representing higher risk in the experimental intervention group.

We calculated mean differences (MDs) and standardised mean differences (SMDs) for our continuous outcomes. For pain assessment, we used the numerical rating scale to measure the

mean difference between groups [37]. Visual analog scale (VAS) (0 to 100) was converted into the numerical rating scale (NRS) (0 to 10) by dividing with 10 [37].

**Assessment of heterogeneity.**   We investigated forest plots to visually assess any sign of heterogeneity. We secondly assessed the presence of statistical heterogeneity by $Chi^2$ test (threshold $P < 0.10$) and measured the quantities of heterogeneity by $I^2$ statistic and tau $(\tau)^2$ statistic [38, 39]. We also investigated heterogeneity through subgroup analyses. Ultimately, we decided whether the assessment of heterogeneity showed that meta-analysis should be avoided [36].

**Assessment of reporting biases.**   We used funnel plots to assess reporting bias, although it should be noted that funnel plots assess small-study effects [36]. Funnel plots were performed if 10 or more trials were included [36]. For dichotomous outcomes, we assessed asymmetry with the Harbord test [40] if $\tau^2$ was less than 0.1 and with the Rücker test if $\tau^2$ was more than 0.1. For continuous outcomes, we used the regression asymmetry test [41].

**Assessment of statistical and clinical significance.**   We performed all meta-analyses using Review Manager 5.4.1 and STATA 16.1 [42, 43]. We assessed our intervention effects with both random-effects meta-analyses (DerSimonian method) [44] and fixed-effect meta-analyses (DeMets method) [43, 45]. We primarily reported the more conservative point estimate of the two and the less conservative result as a sensitivity analysis [46]. To control random errors when analysing our primary outcomes, we adjusted the threshold for statistical significance using the procedure suggested by Jakobsen and colleagues [46]. We used four primary outcomes and therefore considered a P-value of 0.02 as the threshold for statistical significance [46]. When analysing secondary and exploratory outcomes, we considered a P-value of 0.05 as the threshold for statistical significance as these outcomes were considered hypothesis-generating only [46].

We used Trial Sequential Analysis (TSA) to control for the risks of random errors [47]. Traditional meta-analysis runs the risk of random errors due to sparse data and repetitive testing of accumulating data when updating reviews. We wished to control the risks of type I errors and type II errors [46]. By conducting TSA on the outcomes, we could calculate the required information size, i.e., the number of participants needed in a meta-analysis to detect or reject our anticipated intervention effects [46].

For dichotomous outcomes, we estimated the required information size based on the observed proportion of participants with an outcome in the control group, a relative risk reduction of 20% in the experimental group, an alpha of 2.0%, a beta of 10%, and the diversity suggested by the trials in the meta-analysis. For the outcome pain assessment on VAS or NRS, we used an analgesic effect equivalent to 10 mm on VAS or 1 point on NRS. For the outcome 24-hour morphine consumption we used an effect equivalent to at least 5 mg morphine. For the remaining continuous outcomes, we used the observed standard deviation (SD), a mean difference of the observed SD/2, an alpha of 2.0% for our primary and secondary outcomes, a beta of 10%, and the diversity suggested by the trials in the meta-analysis [33]. The above-mentioned intervention effects used in the TSA were also predefined as the minimal important differences (MIDs) of the review [33].

We used a 'best-worst case' and a 'worst-best case' sensitivity analysis to assess the impact of missing data (incomplete outcome data bias) [36]. We used the Grading of Recommendations Assessment, Development and Evaluation (GRADE) to assess the certainty of the evidence [48, 49].

# Results

Our literature search identified 5766 records. After removing duplicates, 4302 records remained. We excluded 4196 records based on title or abstract. We excluded another 41

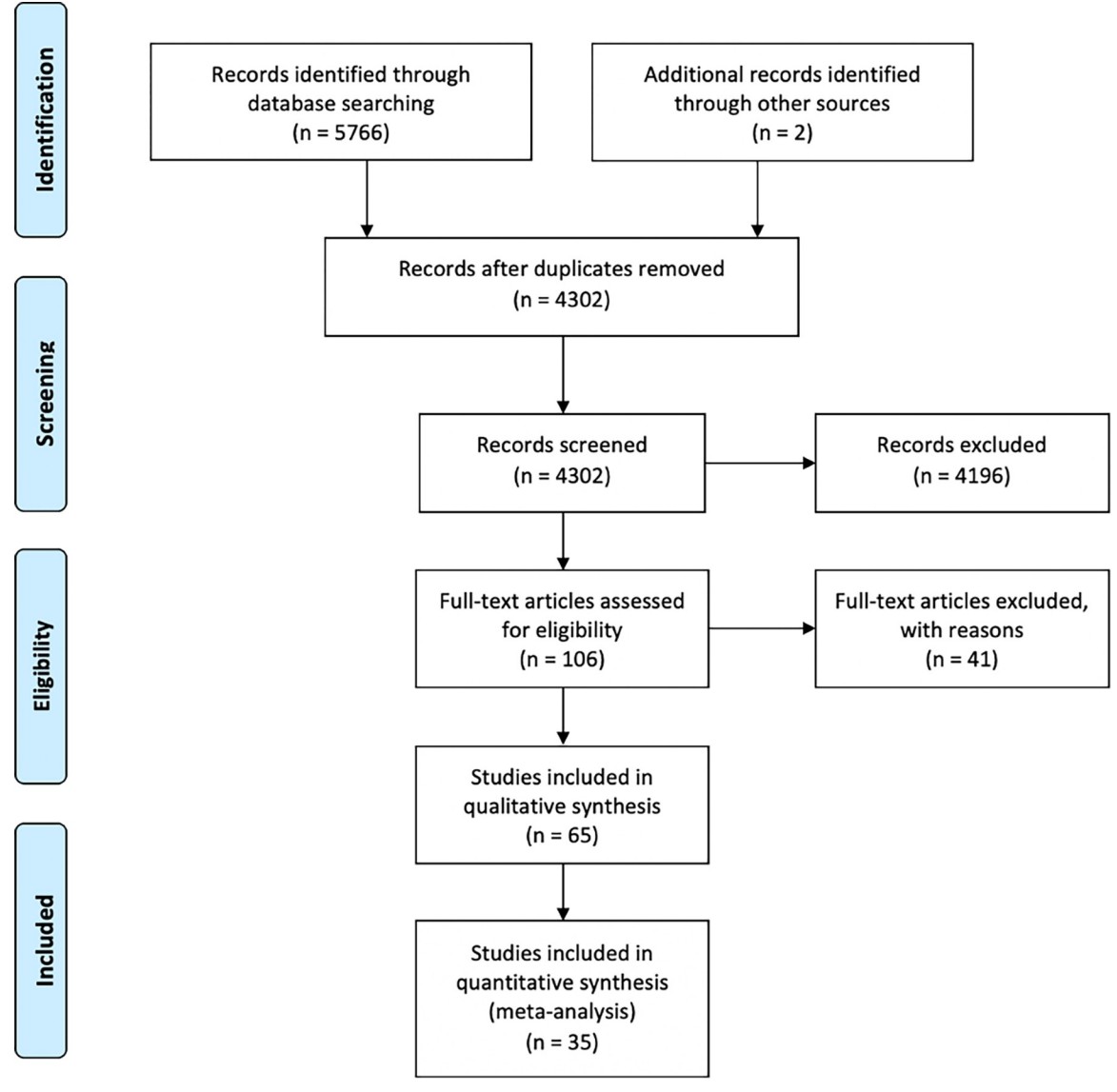

**Fig 1. PRISMA flow diagram.**

records based on the full texts (**Fig 1**). We included a total of 65 clinical trials randomising 7017 participants [50–112].

All 65 trials compared cannabinoids versus placebo. The mean age of included participants was 50.4 years ranging from a mean of 20.7 years [74] to 63.7 years [63] (see **Table 1** for more details).

The participants were included in the trials based on the different pain diagnoses. Forty-four trials randomised 4306 participants with chronic pain (34 trials in neuropathic pain, and 10 trials in chronic nociceptive pain); 9 trials randomised 1681 participants with cancer pain; 10 trials randomised 965 participants with acute pain; and 2 trials randomised 65 participants with fibromyalgia-related pain.

Length of maximum follow-up of the included trials varied between 7 hours [71] and 16 weeks [92] with a mean length of follow-up of 7.3 weeks.

**Table 1. Tables of included randomised clinical trials.**

| Trial | Sample size | Medical condition | Mean age | % male | Type of cannabinoid and administration | Treatment duration |
|---|---|---|---|---|---|---|
| *Abrams 2007* | 55 | HIV-associated sensory neuropathy | 48.5 | 49% | Cannabis cigarettes weighing on average 0.9 g. Active cannabis cigarettes contained 3.56% (smoked) | 5 days |
| *Abrams 2020* | 27 | Chronic pain and episodic acute pain caused by vasoocclusive crises caused by SCD | 37.6 | 47.8% | Cannabis plant material containing 4.4% THC and 4.9% CBD (inhaled via vaporizer) | 5 days |
| *Almog 2020* | 27 | Peripheral neuropathic pain, complex regional pain syndrome (CRPS) | 48.3 | 70.4% | Cannabis flos containing 22% THC, <0.1% cannabidiol (CBD), <0.2% cannabinol (CBN) (inhaled) | 2 days |
| *Aronow 1974* | 10 | Chronic angina | 47.3 | 100% | THC (inhaled) | 4 days |
| *Beaulieu 2006 (1 mg)* | 16 | Acute pain, post operative | 53 | 25% | Nabilone (oral) | 1 day |
| *Beaulieu 2006 (2 mg)* | 14 | Acute pain, post operative | 53 | 21% | Nabilone (oral) | 1 day |
| *Berman 2004* | 48 | Neuropathic pain (Brachial plexus injury) | 39 | 95% | THC/CBD or THC only(oromucosal spray) | 14 days |
| *Bebee 2021* | 100 | Acute, non-traumatic, low back pain | 47 | 56% | Single-dose synthetic oral cannabidiol (400 mg) | Single-dose, 7 hour follow up |
| *Blake 2006* | 58 | Chronic pain (reumatoid arthritis) | 62.8 | 20% | THC/CBD or THC only(oromucosal spray) | 5 weeks |
| *Buggy 2003* | 40 | Acute pain, post operative | 46.3 | 0% | THC (oral) | 1 day |
| *Collin 2010* | 337 | Multiple sclerosis, central neuropathic pain | 47.6 | 38% | THC/CBD (oromucosal spray) | 14 weeks |
| *Colwill 2020* | 70 | Acute pain. Medical abortion associated pain | 28.2 | 0% | Dronabinol (oral capsules) | Single dose |
| *Conte 2009* | 18 | Multiple sclerosis, central neuropathic pain | 51.1 | 66% | THC/CBD (oromucosal spray) | 3 weeks |
| *Corey-bloom 2012* | 30 | Multiple sclerosis, central neuropathic pain | 51 | 37% | THC (smoked) | 3 days |
| *Cote 2016* | 56 | Acute pain, postoperative head and neck cancer | 63.6 | 82% | Nabilone (oral) | 3 weeks |
| *de Vries 2017* | 24 | Chronic pain (chronic pancreatitis) | 51.8 | 62% | THC (oral) | 1 day |
| *de Vries 2016* | 50 | chronic pain (chronic pancreatitis. post surgery pain) | 52 | 50% | THC (oral) | 7 weeks |
| *Eibach 2021* | 32 | HIV-associated sensory neuropathy | 50.3 | 97% | Cannabidivarin (phytocannabinoid derived from cannabis sativa L. plant) (oral) | 4 weeks |
| *Ellis 2009* | 34 | HIV-associated sensory neuropathy | 49.1 | 97% | THC (smoked) | 5 days |
| *Fallon 2017* | 399 | Cancer pain | 59.8 | 51% | THC/CBD (oromucosal spray) | 5 weeks |
| *Fallon 2017* | 206 | Cancer pain | 61.5 | 57% | THC/CBD (oromucosal spray) | 5 weeks |
| *Hunter 2018* | 319 | Osteoarthritis | 62 | | ZYN002 (Transdermal) | 12 weeks |
| *Issa 2014* | 60 | Chronic, non-cancer pain | 43.5 | 46% | Dronabinol (oral capsules) | 3 x single doses Placebo, 10 or 20 mg dronabinol capsules in 1 of 6 randomly allocated sequences. |
| *Jain 1981* | 56 | Acute pain. Severe fracture or trauma pain. post-operative | 28 | 91% | Levonantradol (Intramuscular) | 7 hrs |
| *Jochimsen 1978* | 35 | Cancer pain | 57 | 17% | BPP (oral) | 2 x single dose (2 and 4 mg of BPP) |
| *Johnson 2010 (THC)* | 87 | Cancer pain | 60.2 | 52% | THC (oromucosal spray) | 2 weeks |
| *Johnson 2010 (THC/CBD)* | 90 | Cancer pain | 60.2 | 55% | THC/CBD (oromucosal spray) | 2 weeks |

*(Continued)*

**Table 1.** (Continued)

| Trial | Sample size | Medical condition | Mean age | % male | Type of cannabinoid and administration | Treatment duration |
|---|---|---|---|---|---|---|
| *Kalliomäki 2013* | 120 | Acute post-operative pain. Lower third molar surgery | 20.7 | 100% | AZD1940 (C1+C2 agonist) (oral) | 1 day |
| *Kantor 1981* | 61 | Acute post-operative pain | ? | ? | Levonantradol (parental and oral) | 1 day (3 x single dose) |
| *Karst 2003* | 21 | Chronic, neuropathic pain | 51 | 62% | CT3 (oral capsules) | 1 week |
| *Killestein 2002* | 16 | Multiple sclerosis, central neuropathic pain | 46 | ? | Dronabinol (oral capsules) | 4 weeks |
| *Langford 2013* | 339 | Multiple sclerosis, central neuropathic pain | 49 | 32% | THC/CBD (oromucosal spray) | 14 weeks |
| *Leocani 2014* | 43 | Multiple sclerosis, central neuropathic pain | ? | 53% | THC/CBD (oromucosal spray) | 4 weeks |
| *Levin 2017* | 340 | Acute post-operative pain | 49.8 | 0% | Nabilone (oral) | Single-dose |
| *Lichtman 2018* | 397 | Cancer pain | 59.9 | 54% | THC/CBD (oromucosal spray) | 5 weeks |
| *Lynch 2014* | 18 | Post-chemo neuropathic pain | 56 | 16% | THC/CBD (oromucosal spray) | 4 weeks |
| *Malik 2017* | 19 | Chronic. functional chest pain | 43 | 84% | Dronabinol (oral capsules) | 4 weeks |
| *Marková 2017* | 106 | Multiple sclerosis, central neuropathic pain | 51.3 | 70.2% | THC/CBD (oromucosal spray) | 12 weeks |
| *Narang 2008* | 30 | Chronic non-cancer pain: neuropathic (N = 7). nociceptive (N = 7). mixed neuropathic and nociceptive (N = 11). and uncategorized (N = 5) pain. | 43.5 | 46.7% | Dronabinol (oral capsules) | Single-dose |
| *NCT01606202* | 116 | Central neuropathic pain associated with spinal cord injury | 48.1 | 78.4% | THC/CBD (oromucosal spray) | 3 weeks |
| *Nurmikko 2007* | 125 | Chronic neuropathic pain | 53.3 | 40.8% | THC/CBD (oromucosal spray) | 5 weeks |
| *Ostenfeld 2011 (100 mg)* | 50 | Acute Postoperative pain (molar tooth extraction) | 25.9 | 50% | GW842166 (selective noncannabinoid CB2 agonist) (oral doses) | Single-dose |
| *Ostenfeld 2011 (800 mg)* | 42 | Acute postoperative pain (molar tooth extraction) | 25.9 | 50% | GW842166 (selective noncannabinoid CB2 agonist) (oral doses) | Single-dose |
| *Portenoy 2012 (1–4 sprays)* | 121 | Cancer pain | 58 | 51.7% | THC/CBD (oromucosal spray) | 5 weeks |
| *Portenoy 2012 (11–16 sprays)* | 120 | Cancer pain | 58 | 51.7% | THC/CBD (oromucosal spray) | 5 weeks |
| *Portenoy 2012 (4–10 sprays)* | 119 | Cancer pain | 58 | 51.7% | THC/CBD (oromucosal spray) | 5 weeks |
| *Riva 2016* | 60 | Central neuropathic pain | 57.8 | 57% | THC/CBD (oromucosal spray) | 6 weeks |
| *Rog 2005* | 66 | Multiple sclerosis, central neuropathic pain | 49.2 | 21.2% | THC/CBD (oromucosal spray) | 1 week |
| *Schimrigk 2017* | 240 | Multiple sclerosis, central neuropathic pain | 47.7 | 27.1% | THC/CBD (oromucosal spray) | 16 weeks |
| *Selvarajah 2010* | 297 | Diabetic neuropathy | 59.5 | 61.6% | THC/CBD (oromucosal spray) | 14 weeks |
| *Serpell 2013* | 246 | Perpheral neuropathic pain | 57.3 | 39% | THC/CBD (oromucosal spray) | 15 weeks |
| *Skrabek 2008* | 40 | Fibromyalgia | 49 | 7.5% | Nabilone (oral) | 4 weeks |
| *Stambaugh 1981* | 30 | Cancer pain | ? | ? | Levonantradol (intramuscular) | 4 x single doses (intramuscular Levo 0.5 mg, Levo 2 mg, Morphine 10 mg and placebo) |
| *Staquet 1978* | 30 | Cancer pain | Age range fram 21–75 | ? | Synthetic nitrogen analog of tetrahydrocannabinol (NIB) (oral capsules) | 3 x single doses (codeine, NIB, placebo) |
| *Svendsen 2004* | 24 | Multiple sclerosis. central neuropathic pain | 50 | 45.8% | Dronabinol (oral) | 3 weeks |
| *Toth 2012* | 26 | Diabetic neuropathy | 62 | 45% | Nabilone (oral) | 5 weeks |

*(Continued)*

**Table 1.** (Continued)

| Trial | Sample size | Medical condition | Mean age | % male | Type of cannabinoid and administration | Treatment duration |
|---|---|---|---|---|---|---|
| *Turcott 2018* | 47 | Cancer pain | 57 | 21% | Nabilone (oral) | 8 weeks |
| *Turcotte 2015* | 15 | Multiple sclerosis, central neuropathic pain | 45 | 13% | Nabilone (oral) | 9 weeks |
| *Van de Donk 2019* | 25 | Fibromyalgia | 39 | 0% | Bedrocan 100 mg (22% THC. 1% CBD). Bediol 200 mg (6.3% THC. 8% CBD). Bedrolite 200 mg (<1% THC. 9% CBD) (Vaporised) | 4 x single doses with at least 2 weeks between each dose (including placebo) |
| *van Amerongen 2018* | 24 | Multiple sclerosis, central neuropathic pain | 54 | 33.3% | THC (ECP002A) (oral capsules) | 4 weeks |
| *Vela 2021* | 136 | Chronic pain (hand osteo arthritis and psoriatic arthritis) | 61,75 | 55% | Synthetic CBD 20–30 mg (oral tablet) | 12 weeks |
| *Wade 2003* | 24 | Chronic neuropathic pain | 48 | 50% | Smoked cannabis | 8 weeks (10 weeks including placebo) |
| *Wade 2004* | 160 | Multiple sclerosis, central neuropathic pain | 50.7 | 38% | THC/CBD (oromucosal spray) | 10 weeks |
| *Wallace 2015* | 16 | Diabetic neuropathy | 56.9 | 56% | low (1% tetrahydrocannabinol. THC). medium (4% THC). or high (7% THC) dose of cannabis (Inhaled aerosolized cannabis/ vaporizer) | 4 x Single doses of cannabis (1%, 4%, and 7% THC, placebo), separated by 2 weeks each |
| *Ware 2010* | 23 | Chronic neuropathic pain caused by trauma or surgery | 45.4 | 48% | 2.5%. 6% and 9.4% tetrahydrocannabinol (THC containing cigarettes) | 2 weeks |
| *Wilsey 2008* | 38 | Central and peripheral neuropathic pain | 46 | 52% | High-dose cannabis(7% delta-9-THC), low-dose cannabis (3.5% delta-9-THC), and placebo cigarettes. | 3 x 6 hour experimental sessions with at least 3 days between each session (including placebo) |
| *Wissel 2006* | 13 | Central neuropathic pain | 44.8 | 30% | Nabilone (oral) | 4 weeks |
| *Zajicek 2003* | 330 | Multiple sclerosis, central neuropathic pain | 50 | 34% | Marinol (synthetic delta9-THC) (oral capsules) | 13–14 weeks |
| ***Zajicek 2012*** | 327 | Multiple sclerosis, central neuropathic pain | 50 | 34% | cannador (a cannabis extract. containing delta-9-THC and cannabidiol) (oral) | 12–14 weeks |
| *Zajicek 2012* | 279 | Multiple sclerosis, central neuropathic pain | 51.9 | 36% | THC cannabis extract (oral capsules) | 12 weeks |

Only 35 of the 65 trials provided data that could be included in our meta-analysis. The primary reason that trial data could not be included in the meta-analyses was that the trials were designed as cross-over trials and that they did not provide data at the end of the first phase. These trials are described qualitatively in the paragraph '**Trials not contributing with data in our meta-analyses'.** Fifty-nine of the 65 included trials were at high risk of bias, see '**Supplement 4 in S1 File**' and '**S4 File'** for more information on bias assessment.

## Primary outcomes

**All-cause mortality.** Seven trials randomising 2073 participants reported on all-cause mortality. Meta-analysis showed no evidence of a difference between cannabinoids versus placebo (RR 1.20; 98% CI 0.85 to 1.67; P = 0.22; low certainty of evidence; **S1 Fig in S1 File**). Visual inspection of the forest plot, $I^2$-statistic ($I^2 = 7\%$), and tau$^2$ statistic ($\tau^2 = 0.01$; $\tau = 0.1$) indicated low heterogeneity. TSA showed that there was not enough information to confirm or reject that cannabinoids versus placebo increased the risk of all-cause mortality by 20% or more (**S2 Fig in S1 File**). We assessed this outcome result at high risk of bias, see '**Supplement 4 in S1 File**' and **S3 and S4 Figs in S1 File.**

Test for interaction showed no evidence of a difference when comparing trials at high risk of bias to trials at low risk of bias (P = 0.87) (**S5 Fig in S1 File**); trials at risk of vested interests to trials at no risk of vested interests (P = 0.87) (**S6 Fig in S1 File**); trials assessing different types of pain, i.e., cancer pain and chronic pain (P = 0.43) (**S7 Fig in S1 File**); trials comparing the effects of different types of cannabinoids (P = 0.78) (**S8 Fig in S1 File**). The remaining planned subgroup analyses were not possible to conduct due to lack of relevant data.

**Pain.** Twenty-six trials randomising 4110 participants assessed pain using either VAS (8 trials) or NRS (18 trials). We converted all pain measures to NRS as described in the '**Methods**' section.

The visual inspection of the forest plot and test for subgroup difference (P = 0.02), showed that the effects of cannabinoids seemed to differ between trials randomising participants with acute pain, cancer pain, and chronic pain (**S9 Fig in S1 File**). It was therefore not justifiable to meta-analyse the results of trials including the different types of pain. Hence, we chose to report results separately for each group of trials (acute pain; cancer pain; and chronic pain).

**Acute pain.** Four trials randomising 530 participants suffered from acute pain. Meta-analysis showed no evidence of a difference between cannabinoids versus placebo (mean difference NRS 0.52; 98% CI -0.40 to 1.43; P = 0.19; very low certainty of evidence; **S10 Fig in S1 File**). Visual inspection of the forest plot, $I^2$-statistic ($I^2$ = 90%), and tau$^2$ statistic ($\tau^2$ = 0.67, $\tau$ = 0.82) indicated high heterogeneity that could not be resolved. TSA showed that we had not enough information to confirm or reject that cannabinoids versus placebo reduced acute pain (**S11 Fig in S1 File**). We assessed this outcome result at high risk of bias, see '**Supplement 4 in S1 File**'.

Test of interaction showed evidence of a difference when comparing trials at high risk of bias to trials at low risk of bias (P = 0.035) (**S12 Fig in S1 File**). When trials at low risk of bias and trials at high risk of bias were analysed separately we found no evidence of a difference between cannabinoids versus placebo (**S12 Fig in S1 File**).

Test of interaction showed no evidence of a difference when comparing trials at risk of vested interests to trials at no risk of vested interests (P = 0.25) (**S13 Fig in S1 File**).

All remaining planned subgroup analyses were not possible to conduct due to lack of relevant data.

A post-hoc subgroup analysis comparing trials with long-term follow-up to trials with short-term follow-up showed no evidence of a difference (P = 0.10) (**S14 Fig in S1 File**).

**Cancer pain.** Six trials randomising 1550 participants suffered from cancer pain. Meta-analysis showed no evidence of a difference between cannabinoids versus placebo (mean difference NRS -0.13; 98% CI -0.33 to 0.06; P = 0.1; low certainty of evidence; **S15 Fig in S1 File**). Visual inspection of the forest plot, $I^2$-statistic ($I^2$ = 2%), and tau$^2$ statistic ($\tau^2$ = 0.00) indicated low heterogeneity. TSA showed that there was enough information to reject that cannabinoids versus placebo reduced cancer pain (**S16 Fig in S1 File**). We assessed this outcome result as high risk of bias, see '**Supplement 4 in S1 File**'.

Test of interaction showed no evidence of a difference when comparing trials assessing the effects of different types of cannabinoids (P = 0.71) (**S17 Fig in S1 File**).

All remaining planned subgroup analyses were not possible to conduct due to lack of relevant data.

A post-hoc subgroup analysis comparing trials with long-term follow-up to trials with short-term follow-up showed no evidence of a difference (P = 0.06) (**S18 Fig in S1 File**).

**Chronic pain.** Sixteen trials randomising 2030 participants suffered from chronic pain. Meta-analysis showed that cannabinoids reduced chronic pain, but the effect size was below the predefined MID and the MID was not included in the 98% confidence interval (mean difference NRS -0.43; 98% CI -0.72 to -0.15; P = 0.0004; low certainty of evidence; **Fig 2**, **S19 Fig in S1 File**). Visual inspection of the forest plot, $I^2$-statistic ($I^2$ = 64%), and tau$^2$ statistic ($\tau^2$ = 0.11) indicated moderate heterogeneity that could not be resolved. TSA showed that there was enough

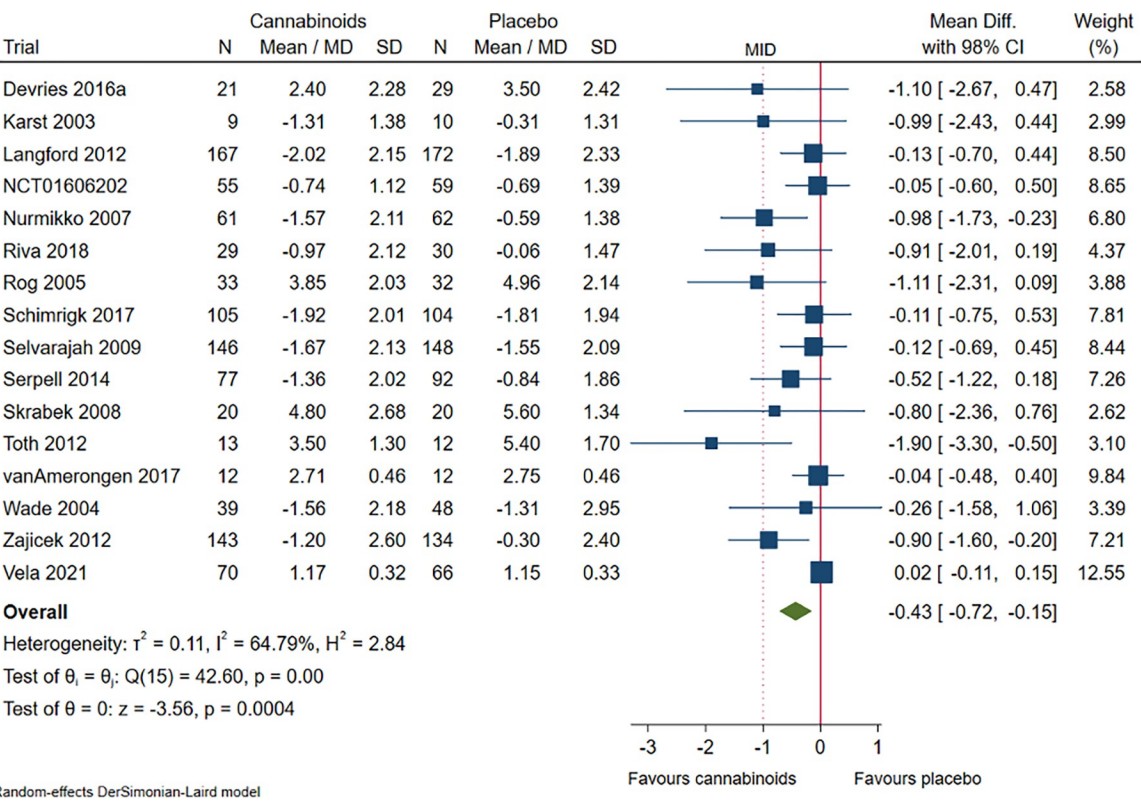

| Trial | Cannabinoids N | Mean / MD | SD | Placebo N | Mean / MD | SD | Mean Diff. with 98% CI | Weight (%) |
|---|---|---|---|---|---|---|---|---|
| Devries 2016a | 21 | 2.40 | 2.28 | 29 | 3.50 | 2.42 | -1.10 [ -2.67, 0.47] | 2.58 |
| Karst 2003 | 9 | -1.31 | 1.38 | 10 | -0.31 | 1.31 | -0.99 [ -2.43, 0.44] | 2.99 |
| Langford 2012 | 167 | -2.02 | 2.15 | 172 | -1.89 | 2.33 | -0.13 [ -0.70, 0.44] | 8.50 |
| NCT01606202 | 55 | -0.74 | 1.12 | 59 | -0.69 | 1.39 | -0.05 [ -0.60, 0.50] | 8.65 |
| Nurmikko 2007 | 61 | -1.57 | 2.11 | 62 | -0.59 | 1.38 | -0.98 [ -1.73, -0.23] | 6.80 |
| Riva 2018 | 29 | -0.97 | 2.12 | 30 | -0.06 | 1.47 | -0.91 [ -2.01, 0.19] | 4.37 |
| Rog 2005 | 33 | 3.85 | 2.03 | 32 | 4.96 | 2.14 | -1.11 [ -2.31, 0.09] | 3.88 |
| Schimrigk 2017 | 105 | -1.92 | 2.01 | 104 | -1.81 | 1.94 | -0.11 [ -0.75, 0.53] | 7.81 |
| Selvarajah 2009 | 146 | -1.67 | 2.13 | 148 | -1.55 | 2.09 | -0.12 [ -0.69, 0.45] | 8.44 |
| Serpell 2014 | 77 | -1.36 | 2.02 | 92 | -0.84 | 1.86 | -0.52 [ -1.22, 0.18] | 7.26 |
| Skrabek 2008 | 20 | 4.80 | 2.68 | 20 | 5.60 | 1.34 | -0.80 [ -2.36, 0.76] | 2.62 |
| Toth 2012 | 13 | 3.50 | 1.30 | 12 | 5.40 | 1.70 | -1.90 [ -3.30, -0.50] | 3.10 |
| vanAmerongen 2017 | 12 | 2.71 | 0.46 | 12 | 2.75 | 0.46 | -0.04 [ -0.48, 0.40] | 9.84 |
| Wade 2004 | 39 | -1.56 | 2.18 | 48 | -1.31 | 2.95 | -0.26 [ -1.58, 1.06] | 3.39 |
| Zajicek 2012 | 143 | -1.20 | 2.60 | 134 | -0.30 | 2.40 | -0.90 [ -1.60, -0.20] | 7.21 |
| Vela 2021 | 70 | 1.17 | 0.32 | 66 | 1.15 | 0.33 | 0.02 [ -0.11, 0.15] | 12.55 |
| **Overall** | | | | | | | -0.43 [ -0.72, -0.15] | |

Heterogeneity: $\tau^2 = 0.11$, $I^2 = 64.79\%$, $H^2 = 2.84$
Test of $\theta_i = \theta_j$: Q(15) = 42.60, p = 0.00
Test of $\theta = 0$: z = -3.56, p = 0.0004

Random-effects DerSimonian-Laird model

**Fig 2. Forest plot of the meta-analysis of chronic pain with 98% CI.**

information to confirm that cannabinoids versus placebo reduced chronic pain with a statistically significant effect (**S20 Fig in S1 File**). We assessed this outcome result as high risk of bias, see '**Supplement 4 in S1 File**'.

Test for interaction showed no evidence of a difference when comparing trials at high risk of bias to trials at low risk of bias (P = 0.31) (**S21 Fig in S1 File**); and trials at risk of vested interests to trials at no risk of vested interests (P = 0.57) (**S22 Fig in S1 File**). Test for interaction showed evidence of a difference when comparing trials assessing different types of chronic pain, i.e., neuropathic pain (including central, peripheral, and mixed), fibromyalgia, and visceral nociceptive pain (P = 0.01) (**S23 Fig in S1 File**); and trials comparing the effects of different types of cannabinoids (P < 0.001) (**S24 Fig in S1 File**).

The funnel plot showed no clear signs of small-study effects (**S25 Fig in S1 File**).

A post-hoc subgroup analysis comparing trials with long-term follow-up to trials with short-term follow-up showed no evidence of a difference (P = 0.59) (**S26 Fig in S1 File**).

**Serious adverse events.** Eighteen trials randomising 3980 participants reported serious adverse events. Meta-analysis showed no evidence of a difference between cannabinoids versus placebo (RR 1.18; 98% CI 0.95 to 1.45; P = 0.07; low certainty of evidence; **Fig 3**, **S27 Fig in S1 File**). Visual inspection of the forest plot, $I^2$-statistic ($I^2 = 0\%$), and $tau^2$ statistic ($\tau^2 = 0.00$) indicated low heterogeneity. TSA showed that there was not enough information to confirm or reject that cannabinoids versus placebo increased the risk of serious adverse events by 20% or more (**S28 Fig in S1 File**). We assessed this outcome result at high risk of bias, see '**Supplement 4 in S1 File**' and **S29 and S30 Figs in S1 File**.

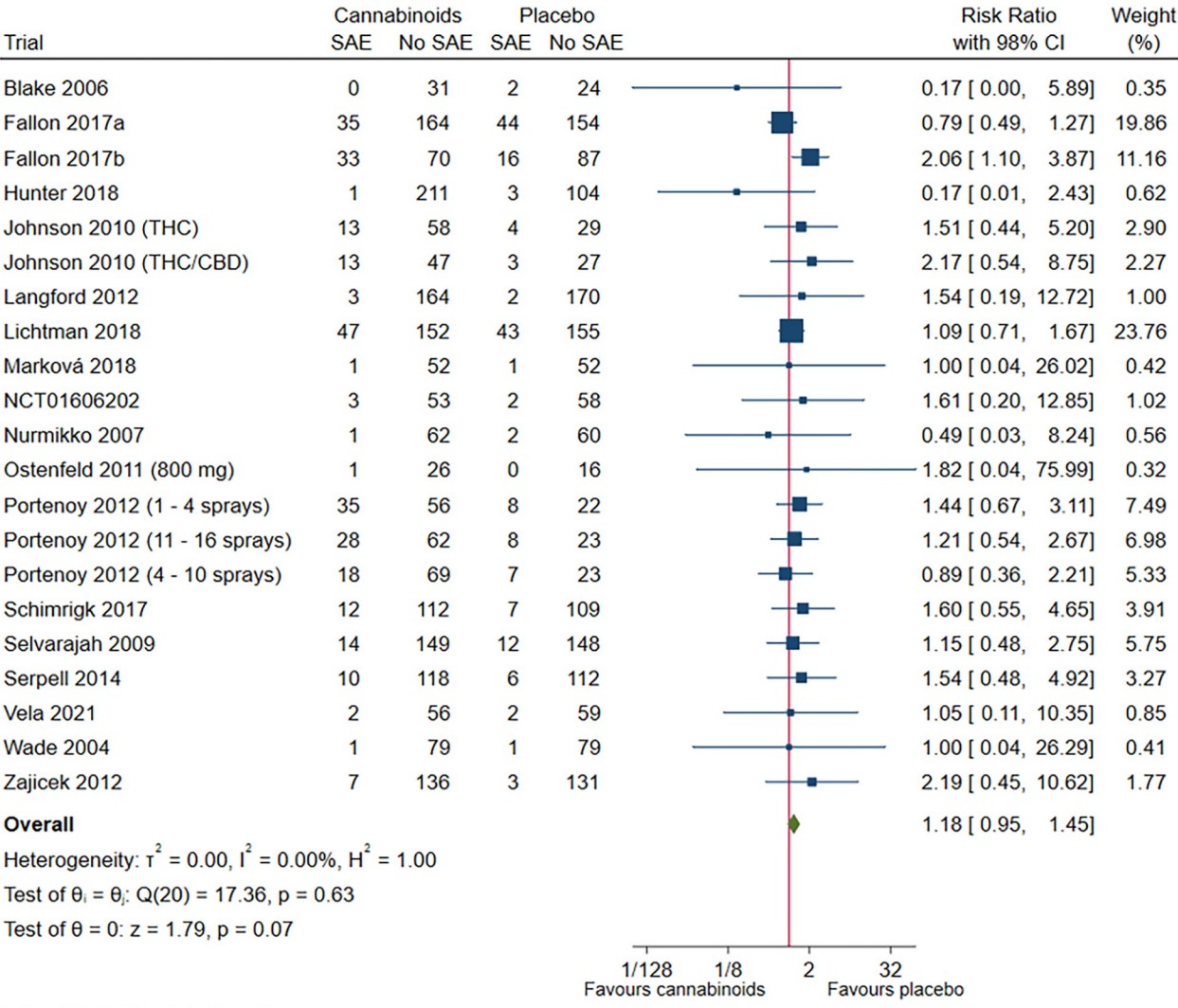

**Fig 3. Forest plot of the meta-analysis of serious adverse events with 98% CI.**

Test for interaction showed no evidence of a difference when comparing trials at risk of vested interests to trials at no risk of vested interests (P = 0.97) (**S31 Fig in S1 File**); trials assessing different types of pain, i.e. acute pain, cancer pain and chronic pain (P = 0.96) (**S32 Fig in S1 File**); trials assessing different types of chronic pain, i.e., neuropathic pain (including central and peripheral) and nociceptive pain (P = 0.16) (**S33 Fig in S1 File**); and trials comparing the effects of different types of cannabinoids (P = 0.79) (**S34 Fig in S1 File**).

The remaining planned subgroup analyses were not possible to conduct due to lack of relevant data.

The funnel plot showed no clear signs of small-study effects (**S35 Fig in S1 File**).

When analysing each specific serious adverse event, cannabinoids did not seem to increase or decrease the risk of any single serious adverse events compared with placebo, see 'S2 File'.

**Quality of life.** Only four trials randomising 548 participants assessed quality of life using either EuroQol 5-D (3 trials) or EORTC-QLQC30 (1 trial). Meta-analysis showed no evidence of a difference between cannabinoids versus placebo (mean difference -1.38; 98% CI -11.8 to

9.04; P = 0.75; very low certainty of evidence; **S36 Fig in** **S1 File**). Visual inspection of the forest plot, $I^2$-statistic ($I^2$ = 86%), and tau$^2$ statistic ($\tau^2$ = 61.98, $\tau$ = 7.87) indicated high heterogeneity which could not be resolved. TSA showed that there was not enough information to confirm or reject that cannabinoids versus placebo improved quality of life (**S37 Fig in** **S1 File**). We assessed this outcome result as high risk of bias, see '**Supplement 4 in** **S1 File**'.

Test for interaction showed no evidence of a difference when comparing trials assessing different types of pain, i.e., cancer pain and chronic pain (P = 0.33) (**S38 Fig in** **S1 File**) and trials comparing the effects of different types of cannabinoids (P = 0.56) (**S39 Fig in** **S1 File**).

All remaining planned subgroup analyses were not possible to conduct due to lack of relevant data.

Meta-analysing the standard mean difference showed no evidence of a difference between cannabinoids versus placebo (SMD -0.19; 95% CI -0.70, 0.32; P = 0.46).

## Secondary outcomes

**Dependence and psychosis.** None of the included randomised clinical trials reported results on the outcomes 'dependence' or 'psychosis' that could be analysed through a meta-analysis.

**Non-serious adverse events.** Twenty-nine trials randomising 5536 participants reported non-serious adverse events. Meta-analysis showed evidence of a harmful effect of cannabinoids (RR 1.20; 95% CI 1.15 to 1.25; P < 0.001; very low certainty of evidence; **Fig 4**, **S40 Fig in** **S1 File**). The number needed to harm (NNH) was seven. Visual inspection of the forest plot, $I^2$-statistic ($I^2$ = 27%), and tau$^2$ statistic ($\tau^2$ = 0.00) indicated low heterogeneity. TSA showed that there was enough information to confirm that cannabinoids versus placebo increased the risk of non-serious adverse events by 20% or more (**S41 Fig in** **S1 File**). We assessed this outcome result at high risk of bias, see '**Supplement 4 in** **S1 File**' and **S42 and S43 Figs in** **S1 File**.

The funnel plot showed signs of small-study effects (**S44 Fig in** **S1 File**).

When analysing each specific non-serious adverse event, cannabinoids seemed to increase the risk of five non-serious adverse events versus placebo, see '**S3 File**'. These were 'dizziness', 'fatigue', 'vertigo', 'nervous system disorders', and 'gastrointestinal disorder'.

Cannabinoids did not seem to decrease the risk of any non-serious adverse events.

**Quality of sleep.** Seventeen trials randomising 3291 participants assessed quality of sleep using numerical rating scale (NRS). Meta-analysis showed that cannabinoids versus placebo improved quality of sleep, but the effect size was below the predefined MID and the MID was excluded in the confidence interval (mean difference NRS -0.42; 95% CI -0.65 to -0.20; P = 0.0003; low certainty of evidence; **Fig 5**, **S45 Fig in** **S1 File**). Visual inspection of the forest plot, $I^2$-statistic ($I^2$ = 74%), and tau$^2$ statistic ($\tau^2$ = 0.15) indicated moderate heterogeneity that could not be resolved. TSA showed that there was enough information to confirm that cannabinoids versus placebo improved quality of sleep with a statistically significant effect (**S46 Fig in** **S1 File**). We assessed this outcome result at high risk of bias, see '**Supplement 4 in** **S1 File**'.

The funnel plot showed no clear signs of small-study effects (**S47 Fig in** **S1 File**).

## Exploratory outcomes

**Twenty-four hours morphine consumption.** Six trials randomising 1546 participants assessed 24-hours morphine consumption. Meta-analysis showed no evidence of a difference between cannabinoids versus placebo (mean difference -0.01; 95% CI -1.60 to 1.59; P = 0.99; low certainty of evidence; **S48 Fig in** **S1 File**). Visual inspection of the forest plot and $I^2$-statistic ($I^2$ = 21%) and tau$^2$ statistic ($\tau^2$ = 29.78, $\tau$ = 5.46) indicated low heterogeneity. TSA showed that there was enough information to reject that cannabinoids versus placebo reduced

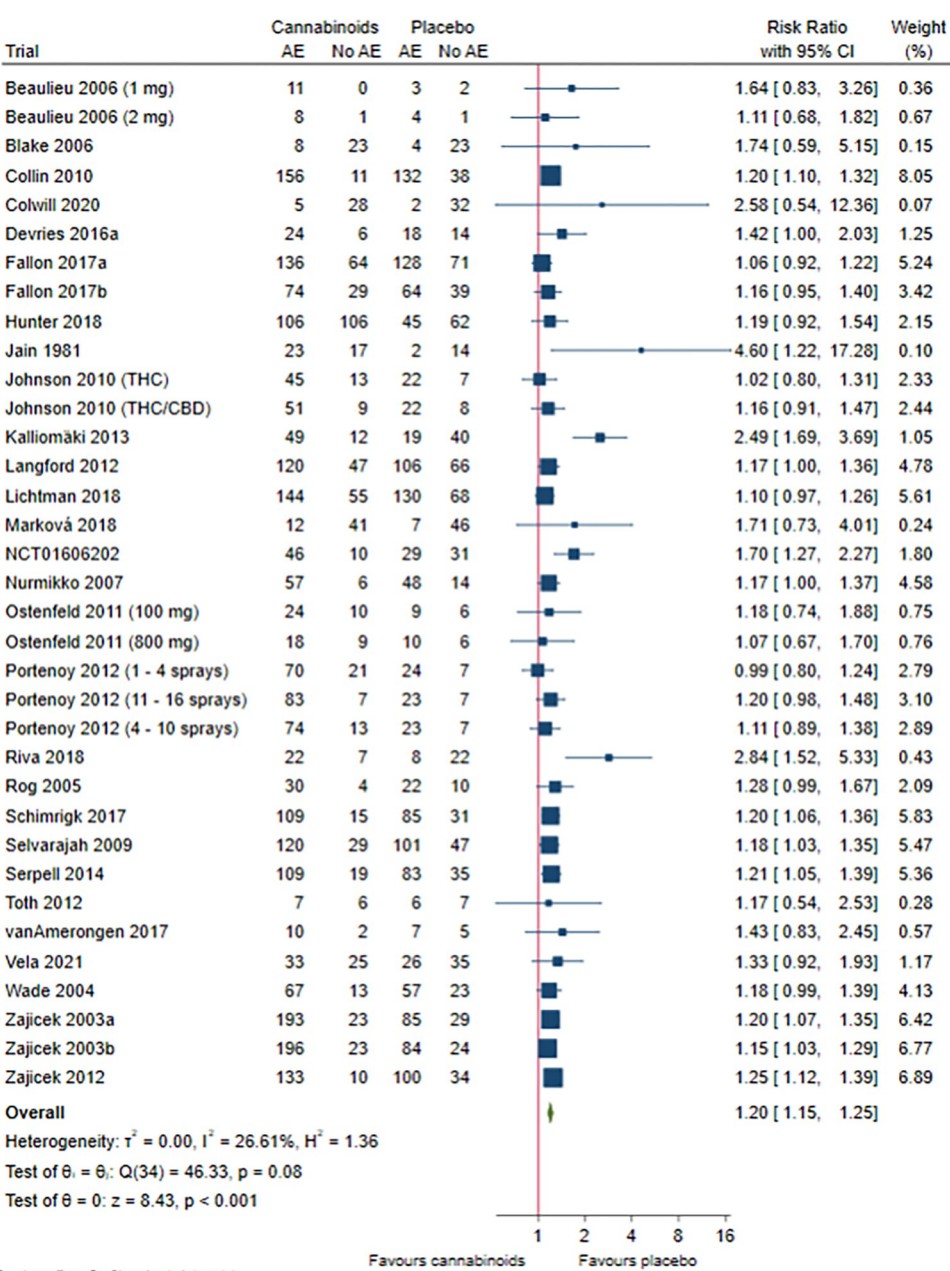

**Fig 4. Forest plot of the meta-analysis of non-serious adverse events with 95% CI.**

24-hours morphine consumption by five mg morphine equivalents or more (**S49 Fig in S1 File**). We assessed this outcome result at high risk of bias, see '**Supplement 4 in S1 File**'.

**Physical function (measured by activities of daily living).** Three trials randomising 320 participants assessed ability to perform activities of daily living (ADL) using Barthel Index. Meta-analysis showed no evidence of a difference between cannabinoids versus placebo (mean difference 0.21; 95% CI -0.10 to 0.51; P = 0.18; very low certainty of evidence; **S50 Fig in S1 File**). Visual inspection of the forest plot and $I^2$-statistic ($I^2 = 0\%$) and tau$^2$ statistic ($\tau^2 = 0.00$) indicated low heterogeneity. TSA could not be performed due to inadequate information size. We assessed this outcome result at high risk of bias, see '**Supplement 4 in S1 File**'.

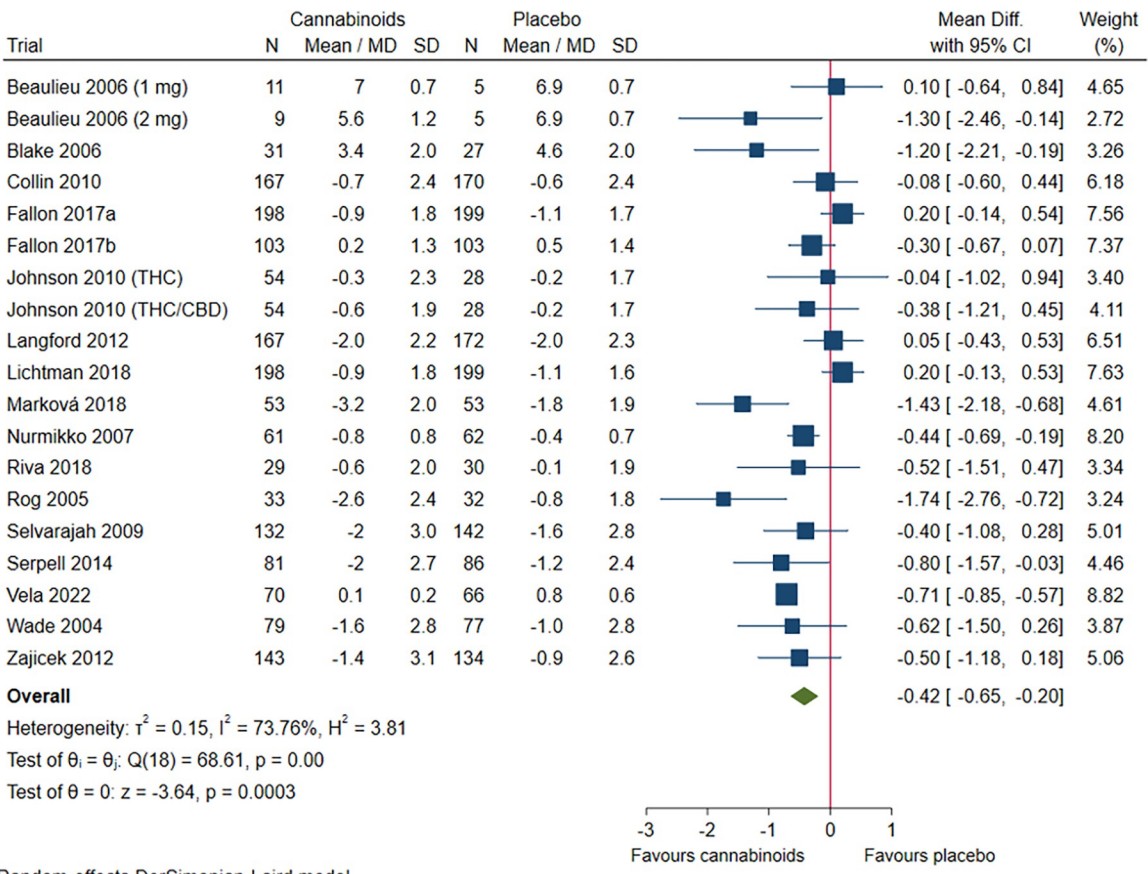

**Fig 5. Forest plot of the meta-analysis of quality of sleep with 95% CI.**

**Depressive symptoms.** Five trials randomising 651 participants assessed depressive symptoms using either HADS-D (3 trials), MADRS (1 trial) or BDI-II (1 trial). Meta-analysis showed no evidence of a difference between cannabinoids versus placebo (mean difference -0.03; 95% CI -0.12 to 0.06; P = 0.49; low certainty of evidence; **S51 Fig in S1 File**). Visual inspection of the forest plot and $I^2$-statistic ($I^2 = 0\%$) and $tau^2$ statistic ($\tau^2 = 0.00$) indicated low heterogeneity. TSA showed that there was enough information to reject that cannabinoids versus placebo improved depressive symptoms (**S52 Fig in S1 File**). We assessed this outcome result at high risk of bias, see '**Supplement 4 in S1 File**'.

**Trials not contributing with data in our meta-analyses.** Nineteen trials randomising 603 participants concluded in favor of an analgesic effect of cannabinoids compared with placebo [50, 52, 62, 67, 70, 71, 75, 82, 83, 85, 96–98, 101, 105, 107–110]. Fifteen trials assessed chronic pain including neuropathic pain (13/15) [50, 52, 62, 67, 82, 85, 98, 101, 105, 107–110] and visceral nociceptive pain (2/15) [70, 83], two trials assessed acute pain [71, 75], and two trials assessed cancer pain [96, 97]. Six trials assessed the use of plant-derived THC extract [50, 52, 62, 107–109], ten trials assessed the use of synthetic THC [70, 71, 75, 83, 85, 96–98, 101, 110], and three trials assessed plant-derived THC/CBD extract [67, 82, 105]. Cannabinoids were generally assessed as safe [50, 52, 62, 67, 82, 98, 101, 105, 108, 109]. Only one trial found a beneficial effect of cannabinoids on quality of life compared with placebo [98]. Mortality was not assessed by any of the trials.

Eleven trials randomising 391 participants concluded against an analgesic effect of cannabinoids compared with placebo [51, 53, 55, 56, 61, 63, 65, 66, 72, 77, 103]. Eight trials assessed chronic including neuropathic pain (4/8) [56, 61, 66, 77], visceral nociceptive pain (3/8) [51, 53, 65], and fibromyalgia (1/8) [103], two trials assessed acute pain [55, 63], and one trial assessed cancer pain [72]. Two trials assessed the use of plant-derived THC extract [53, 66], four trials assessed the use of synthetic THC [63, 65, 72, 77], one trial assessed the use of CBD [55] and four trials assessed plant-derived THC/CBD extract [51, 56, 61, 103]. Cannabinoids were generally assessed as safe [51, 56, 63, 65, 66, 77], however, cannabinoids were not assessed in regards to mortality or quality of life.

## Discussion

The objective of this review was to assess the beneficial and harmful effects of cannabinoids in any dose, formulation, and duration versus placebo or no intervention for participants with any type of pain.

Meta-analysis and TSA showed no evidence of a difference between cannabinoids versus placebo on all-cause mortality, serious adverse events, and quality of life. Meta-analyses showed that cannabinoids compared with placebo reduced chronic pain (in particular central and peripheral neuropathic pain) and improved quality of sleep, but both effect sizes were below our predefined minimal important differences and the minimal important differences were excluded by both confidence intervals. Meta-analyses and TSA showed that cannabinoids increased the risks of non-serious adverse events, which corresponds to a number needed to harm of seven. None of the included trials reported on cannabinoid dependence or psychosis.

According to the European Medicines Agency, P-values are of limited value as relative or absolute differences in terms of adverse effects [113]. A non-significant difference between treatments will not allow for a conclusion on the absence of a difference in safety [113]. In cases of adjustment for multiplicity, the European Medicines Agency state that this can be counterproductive for considerations of safety [113]. Hence, even though meta-analysis and TSA did not show a statistically significant difference when assessing serious adverse events, our results show indications of a harmful effect which should be considered before prescribing cannabinoids for pain.

Our findings are in contrast to the majority of previous reviews which have indicated an adequate analgesic effect of cannabinoids and supported the use of cannabinoids for the treatment of chronic pain [23–26, 114, 115]. Our findings suggest that cannabinoids reduced chronic pain when compared with placebo, however, whether this effect is clinically important seems questionable.

Our findings are in agreement with position statement of the International Association for the Study of Pain (IASP) [116]. IASP have identified important research gaps and due to the lack of high- quality clinical evidence IASP does not currently endorse general use of cannabis and cannabinoids for pain relief [116].

Our findings regarding cancer pain and acute pain are in line with previous reviews on this topic [117–119].

### Strengths

Our present review has several strengths. Our methodology was predefined and was described in detail in our published protocol [33]. To control the risk of random errors, we used TSA [47] and adjusted our thresholds for statistical significance [46]. We included more trials than any other previous review, which has increased our power and precision and strengthened our analyses. We assessed the risk of bias of all included trials to assess the risks of systematic errors

[48, 49], and we used our eight-step procedure to assess if the thresholds for statistical and clinical significance were crossed [46]. Furthermore, we predefined MIDs for all outcomes to assess the clinical implications for patients of our results [33]. Pain level thresholds for acute pain and for chronic pain were predefined based on Olsen et al. [120, 121]. The MID of one point on NRS is considered lenient in comparison to previous reviews, and our predefined lenient quantification decreases the risk of erroneous rejections of cannabinoids' beneficial effects on pain [23, 29]. We were also in contact with several relevant patient associations in Denmark at the protocol stage to select the most relevant patient-important outcomes [33].

## Limitations

Our review also has several limitations. All trials except five [60, 80, 90, 111] were at high risk of bias. Therefore, there is a high risk that our results overestimate the beneficial effects and underestimate the harmful effects of cannabinoids [122–128]. We decided to combine the VAS scores and the NRS scores by converting the results to NRS scores. Even though the two scoring systems correspond very well, some information may be lost in the conversion [129]. Furthermore, a limitation of this systematic review is that we only compared the cannabinoid intervention with placebo. Hence, our results do not assess the effects of cannabinoids compared with other types of analgesics (e.g., opioids).

When conducting a subgroup analysis, one is at risk of study-level confounding because the subgroup analyses are based on aggregate data on a trial level and we did not obtain any individual patient data which may decrease the statistical power of our subgroup analyses. This is a common potential limitation in a meta-analysis of aggregate data.

## Minimal important difference

Pain and quality of sleep measurements are subjective measures, why imprecision could be present when assessing these outcomes. We, therefore, need to be careful before dismissing such outcome results as clinically unimportant. Nevertheless, any outcome result should be related to a predefined minimal important difference (MID) to ensure the scientific validity of trial results and to avoid focus on statistically significant results without importance to patients. If a large number of trial participants are randomised, small and clinically irrelevant intervention effects may lead to statistically significant results and rejection of the null hypothesis [130]. Jaeschke et al. defined the MID as 'the smallest difference in score in the domain of interest which patients perceive as beneficial' [131]. Olsen et al. have conducted two systematic reviews on this matter in order to gather the evidence and present an estimate of the MID for pain [120, 121]. Olsen et al. conducted a systematic review on the MID in patients with acute pain and concluded that the median of the studies' results was 17 mm on VAS (IQR 14 mm to 23 mm) [120]. Another systematic review conducted by Olsen et al. was on the MID in patients with chronic pain and the results showed a median of 23 mm on VAS (IQR 12 mm to 39 mm) when using the within-patient anchor-based method, while the median in studies using the sensitivity-based and specificity-based method was 20 mm on VAS (IQR 15 mm to 30 mm) [121].

Our MID for quality of sleep was not based on previous studies, because such studies have not been conducted. We, therefore, based our estimation of MID on Cohen's D higher than 0.5 [132]. The 0.5 SMD threshold was originally proposed by Cohen (as a minimum for a 'moderate' effect) and has been used as a MID in several studies across medical specialties [132]. Nevertheless, it has to be considered that the MID estimation for quality of sleep compared to the MID for pain is more unclear because studies assessing the MID are lacking.

Several countries have recently either expanded or introduced the medical use of cannabinoids. Of the many different approaches introduced globally only a few have been presented

by the European Monitoring Centre for Drugs and Drug Addiction (EMCDDA) in their report [133]. Canada was one of the first countries to establish a national programme for the medical use of cannabinoids in 1999 and have since evolved to an expanded access programme [134]. New legislation in 2014 licensed more cannabis producers, allowed doctors greater latitude in prescribing, removed federal oversight of prescribing, and permitted patients to receive cannabis directly from licensed producers [135]. Similarly, numerous European countries have allowed the usage of cannabinoids for medical purposes [133]. The different national regulatory frameworks are complicated, but the most common initial approach implemented is to use some form of special access scheme. Examples of countries that have established some form of exceptional use programme or access programme to allow access to cannabinoid preparations for the treatment of pain are Croatia, Denmark, Finland, Norway, Poland, Sweden, Czechia, Germany, Italy, and The Netherlands [133]. According to our present results, the medical use of cannabinoids for pain ought to be reconsidered.

The NASEM report from 2017 on the health effects of marijuana reviewed the evidence on the use of cannabinoids as a treatment for chronic pain [136]. The primary source of information for this summary was based on the works of Whiting et al. suggesting that cannabinoids demonstrate a modest effect on pain [114]. Our present review represents the most comprehensive systematic assessment of the effects of cannabinoids on pain. This systematic review will furthermore work as recommendation for where focus needs to be in future randomised clinical trials.

## Conclusions

Cannabinoids reduced chronic pain and improved quality of sleep, but the effect sizes are of questionable importance. Cannabinoids had no effects on acute pain or cancer pain and increased the risk of non-serious adverse events. The harmful effects of cannabinoids for pain seem to outweigh the potential benefits. The expanded medical use of cannabinoids for pain is at this point questionable.

## Supporting information

**S1 Checklist. PRISMA 2009 checklist.**
(DOC)

**S1 File.**
(DOCX)

**S2 File. Each serious adverse event separately.**
(XLSM)

**S3 File. Each adverse event not considered serious separately.**
(XLSM)

**S4 File. ROB assessment and characteristics of included studies.**
(DOCX)

## Author Contributions

**Conceptualization:** Jehad Barakji, Christian Gluud, Janus Christian Jakobsen.

**Data curation:** Jehad Barakji, Steven Kwasi Korang, Joshua Feinberg, Mathias Maagaard.

**Formal analysis:** Jehad Barakji, Steven Kwasi Korang, Joshua Feinberg, Mathias Maagaard.

**Funding acquisition:** Jehad Barakji.

**Investigation:** Jehad Barakji, Janus Christian Jakobsen.

**Methodology:** Jehad Barakji, Ole Mathiesen, Christian Gluud, Janus Christian Jakobsen.

**Project administration:** Jehad Barakji, Janus Christian Jakobsen.

**Software:** Jehad Barakji, Steven Kwasi Korang.

**Supervision:** Christian Gluud, Janus Christian Jakobsen.

**Validation:** Steven Kwasi Korang, Ole Mathiesen, Christian Gluud, Janus Christian Jakobsen.

**Visualization:** Jehad Barakji, Christian Gluud, Janus Christian Jakobsen.

**Writing – original draft:** Jehad Barakji.

**Writing – review & editing:** Steven Kwasi Korang, Joshua Feinberg, Mathias Maagaard, Ole Mathiesen, Christian Gluud, Janus Christian Jakobsen.

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
