## [Decision Letter · Decision Letter 0]

16 Jun 2022

PONE-D-22-10156Cannabinoids versus placebo for pain: a systematic review with meta-analysis and Trial Sequential AnalysisPLOS ONE

Dear Dr. Barakji,

Thank you for submitting your manuscript to PLOS ONE. After careful consideration, we feel that it has merit but does not fully meet PLOS ONE’s publication criteria as it currently stands. Therefore, we invite you to submit a revised version of the manuscript that addresses the points raised during the review process.

We look forward to receiving your revised manuscript.

Kind regards,

Tariq Jamal Siddiqi

Academic Editor

PLOS ONE

Journal Requirements:

2. Please ensure that you refer to Figures 2-5 in your text as, if accepted, production will need this reference to link the reader to the figure.

3. Please include your tables as part of your main manuscript and remove the individual files. Please note that supplementary tables (should remain/ be uploaded) as separate "supporting information" files.

5.  We noticed you have some minor occurrence of overlapping text with the following previous publication(s), which needs to be addressed:

- https://bmjopen.bmj.com/content/9/10/e031574

In your revision ensure you cite all your sources (including your own works), and quote or rephrase any duplicated text outside the methods section. Further consideration is dependent on these concerns being addressed.

Reviewers' comments:

Reviewer's Responses to Questions

**Comments to the Author**

1. Is the manuscript technically sound, and do the data support the conclusions?

Reviewer #1: Yes

2. Has the statistical analysis been performed appropriately and rigorously? 

Reviewer #1: Yes

3. Have the authors made all data underlying the findings in their manuscript fully available?

Reviewer #1: Yes

4. Is the manuscript presented in an intelligible fashion and written in standard English?

Reviewer #1: No

5. Review Comments to the Author

Reviewer #1: 1. Don’t use short forms like TSA directly in the abstract.

2. Introduction: Please improve upon the Introduction. There should be use of a variety of sentence structures, if possible. Importantly, the gaps in current literature and consequently the rationale for this review remain unclear and should be elaborated upon further in the Introduction. Secondly, please discuss both acute and chronic pain while briefly outlining the comparison of cannabinoids with other relevant treatment therapies for pain. Thirdly, briefly outline the please discuss current guideline recommendations, if any. You may find the following paper useful: https://www.cadth.ca/sites/default/files/pdf/htis/2019/RC1153%20Cannabis%20Chronic%20Pain%20Final.pdf

3. Methods: Please cite “visual analogue scale (VAS) or numerical rating scale (NRS)”. Additionally, please elaborate upon the use of 95% and 98% confidence intervals for dichotomous outcomes – “We calculated risk ratios (RRs) with 95% and 98% confidence intervals (CIs) for our dichotomous outcomes”.

4. Results: The detailed analysis of the data at hand is commendable. Please state an insignificant p value as there being no difference between the cannabinoid and placebo groups, for instance for “Meta-analysis indicated that cannabinoids increased the risk of serious adverse events (RR 1.18; 98% CI 0.95 to 1.45; P = 0.07; low certainty of evidence; Supplementary figure 27).”

5. Use the short form for “Trial Sequential Analysis” as TSA after mentioning the full-form once initially. Also, state the full form of GRADE.

6. Discussion: Please discuss, compare, and contrast findings of other studies with your findings instead of merely stating “Our findings are in contrast to the majority of previous reviews which have indicated an adequate analgesic effect of cannabinoids and supported the use of cannabinoids for the treatment of chronic pain. 20-23 111 112 Our findings suggest that cannabinoids reduced chronic pain when compared with placebo, however, whether this effect is clinically important seems questionable. Our findings regarding cancer pain and acute pain are in line with previous reviews on this topic.113- 115”. Also, improve upon the structure and quality of the Discussion section in general.

7. References: Avoid the use of references more than 10 years old.

6. PLOS authors have the option to publish the peer review history of their article (what does this mean?). If published, this will include your full peer review and any attached files.

Reviewer #1: **Yes: **Warda Ahmed

---

## [Author Response · Author response to Decision Letter 0]

14 Jul 2022

This has been addressed. We have changed the reference style. The headings have been changed according to the PLOS ONE requirements. Figure and table caption have been inserted directly after they have been cited in the text.

2. Please ensure that you refer to Figures 2-5 in your text as, if accepted, production will need this reference to link the reader to the figure.

Figures 2-5 are now referred to in the text.

3. Please include your tables as part of your main manuscript and remove the individual files. Please note that supplementary tables (should remain/ be uploaded) as separate "supporting information" files.

We have now included the “Table 1: Table of included studies” in our manuscript. 

We have now included a paragraph of our supporting information files at the end of the manuscript. This includes captions of all the supplemental files. 

5. We noticed you have some minor occurrence of overlapping text with the following previous publication(s), which needs to be addressed:

- https://bmjopen.bmj.com/content/9/10/e031574

We have now rephrased and changes the sentence structure of the introduction section in order to avoid overlapping text with the above mentioned publication.

In your revision ensure you cite all your sources (including your own works), and quote or rephrase any duplicated text outside the methods section. Further consideration is dependent on these concerns being addressed.

We have carefully checked that all sources are cited and that there is no duplicated text outside the methods section.

Reviewer response:

1.Don’t use short forms like TSA directly in the abstract.

We have changed this accordingly

2. Introduction: Please improve upon the Introduction. There should be use of a variety of sentence structures, if possible. Importantly, the gaps in current literature and consequently the rationale for this review remain unclear and should be elaborated upon further in the Introduction. Secondly, please discuss both acute and chronic pain while briefly outlining the comparison of cannabinoids with other relevant treatment therapies for pain. Thirdly, briefly outline the please discuss current guideline recommendations, if any. You may find the following paper useful: https://www.cadth.ca/sites/default/files/pdf/htis/2019/RC1153%20Cannabis%20Chronic%20Pain%20Final.pdf

The introduction has now been revised and separated into different sections to provide a better overview. We have tried to describe the difference between acute pain, chronic pain and cancer pain more in depth while trying to included current guideline recommendations. We thank the reviewer for this suggestion.

We have also tried to clarify the rational for this systematic review further in the end of the introduction section.

3. Methods: Please cite “visual analogue scale (VAS) or numerical rating scale (NRS)”. Additionally, please elaborate upon the use of 95% and 98% confidence intervals for dichotomous outcomes – “We calculated risk ratios (RRs) with 95% and 98% confidence intervals (CIs) for our dichotomous outcomes”.

We have now cited the visual analogue scale (VAS) or numerical rating scale (NRS) in the methods section. 

We have also elaborated upon the use of two different confidence intervals. This rationale has also been described in the section “Assessment of statistical and clinical significance" previously, therefore we have also referred to this section. 

4. Results: The detailed analysis of the data at hand is commendable. Please state an insignificant p value as there being no difference between the cannabinoid and placebo groups, for instance for “Meta-analysis indicated that cannabinoids increased the risk of serious adverse events (RR 1.18; 98% CI 0.95 to 1.45; P = 0.07; low certainty of evidence; Supplementary figure 27).”

We have changed this to “Meta-analysis showed no evidence of a difference between cannabinoids versus placebo (RR 1.18; 98% CI 0.95 to 1.45; P = 0.07; low certainty of evidence; Fig 3, Supplementary figure 27)”.

5. Use the short form for “Trial Sequential Analysis” as TSA after mentioning the full-form once initially. Also, state the full form of GRADE.

We have changed this accordingly.

6. Discussion: Please discuss, compare, and contrast findings of other studies with your findings instead of merely stating “Our findings are in contrast to the majority of previous reviews which have indicated an adequate analgesic effect of cannabinoids and supported the use of cannabinoids for the treatment of chronic pain. 20-23 111 112 Our findings suggest that cannabinoids reduced chronic pain when compared with placebo, however, whether this effect is clinically important seems questionable. Our findings regarding cancer pain and acute pain are in line with previous reviews on this topic.113- 115”. Also, improve upon the structure and quality of the Discussion section in general.

We have included the position paper of the IASP and compared our results to the findings of the IASP. Further, we have divided the discussion section into several parts to improve the structure. Therefore the following sections appear in the discussion section: “Strengths”, “Limitations” and “Minimal important difference”.

7. References: Avoid the use of references more than 10 years old.

We thank the peer reviewer for this advice. This has been taken into consideration.

---

## [Decision Letter · Decision Letter 1]

17 Aug 2022

Cannabinoids versus placebo for pain: a systematic review with meta-analysis and Trial Sequential Analysis

PONE-D-22-10156R1

Dear Dr. Barakji,

We’re pleased to inform you that your manuscript has been judged scientifically suitable for publication and will be formally accepted for publication once it meets all outstanding technical requirements.

Kind regards,

Tariq Jamal Siddiqi

Academic Editor

PLOS ONE

Additional Editor Comments (optional):

Reviewers' comments:

Reviewer's Responses to Questions

**Comments to the Author**

1. If the authors have adequately addressed your comments raised in a previous round of review and you feel that this manuscript is now acceptable for publication, you may indicate that here to bypass the “Comments to the Author” section, enter your conflict of interest statement in the “Confidential to Editor” section, and submit your "Accept" recommendation.

Reviewer #2: All comments have been addressed

2. Is the manuscript technically sound, and do the data support the conclusions?

Reviewer #2: Yes

3. Has the statistical analysis been performed appropriately and rigorously? 

Reviewer #2: Yes

4. Have the authors made all data underlying the findings in their manuscript fully available?

Reviewer #2: Yes

5. Is the manuscript presented in an intelligible fashion and written in standard English?

Reviewer #2: Yes

6. Review Comments to the Author

Reviewer #2: (No Response)

7. PLOS authors have the option to publish the peer review history of their article (what does this mean?). If published, this will include your full peer review and any attached files.

Reviewer #2: No

---

## [Editor Report · Acceptance letter]

19 Aug 2022

PONE-D-22-10156R1 

Cannabinoids versus placebo for pain: a systematic review with meta-analysis and Trial Sequential Analysis 

Dear Dr. Barakji:

I'm pleased to inform you that your manuscript has been deemed suitable for publication in PLOS ONE. Congratulations! Your manuscript is now with our production department. 

Kind regards, 

on behalf of

Dr. Tariq Jamal Siddiqi 

Academic Editor

PLOS ONE